# Intervertebral Disc Degeneration and Regeneration: New Molecular Mechanisms and Therapeutics: Obstacles and Potential Breakthrough Technologies

**DOI:** 10.3390/cells13242103

**Published:** 2024-12-19

**Authors:** William Taylor, William Mark Erwin

**Affiliations:** 1Department of Surgery, Division of Neurosurgery, University of California at San Diego, 9350 Campus Point Dr., La Jolla, CA 92037, USA; wtaylor@health.ucsd.edu; 2Department of Surgery, Divisions of Orthopaedic and Neurosurgery, University of Toronto, 661 University Ave., Suite 13-1387, Toronto, ON M5G 0B7, Canada

**Keywords:** degenerative disc disease, molecular therapy, notochordal cells, imaging, disc pain

## Abstract

Pain and disability secondary to degenerative disc disease continue to burden the healthcare system, creating an urgent need for effective, disease-modifying therapies. Contemporary research has identified potential therapies that include protein-, cellular- and/or matrix-related approaches; however, none have yet achieved a meaningful clinical impact. The tissue-specific realities of the intervertebral disc create considerable therapeutic challenges due to the disc’s location, compartmentalization, hypovascularization and delicate physiological environment. Furthermore, the imaging modalities currently used in practice are largely unable to accurately identify sources of pain ostensibly discogenic in origin. These obstacles are considerable; however, recent research has begun to shed light on possible breakthrough technologies. Such breakthroughs include revolutionary imaging to better identify tissue sources of pain. Furthermore, novel molecular therapies have been shown to be able to mediate the progression of degenerative disc disease in some large animal studies, and even provide some insight into suppressing the development of tissue sources of discogenic pain. These potential breakthrough technologies have yet to be translated for clinical use.

## 1. Introduction

Degeneration of the intervertebral disc (IVD) is a well-recognized malady that is associated with tremendous healthcare costs and represents the largest single cause of disability worldwide [1]. Degeneration of the intervertebral disc is well characterized by imaging such as CT scanning, MRI and plain-film X-ray. Progressive degeneration, as determined by MRI imaging, is strongly associated with low back pain [2]. Due to the prevalence of symptomatic degenerative disc disease (DDD), significant costs, and a lack of effective treatment, there is an urgency to develop an effective treatment as opposed to temporary symptomatic relief [3]. The IVD is a unique tissue compartment that is mostly avascular (apart from the very periphery of the annulus fibrosis), with the cells of the hypoxic IVD nucleus pulposus (NP) largely dependent on diffusion from the bone marrow of the adjacent vertebral body through the vertebral endplate for its supply of nutrients and exchange of gases [4]. This avascular tissue compartment presents a critical environment for the cells within the disc and compromised capillary buds within the vertebral endplate are thought to limit the supply of nutrients and gases—a phenomenon that may figure prominently in the development and progress of degeneration [5,6]. These biological realities figure prominently in the degenerative cascade of events that is typical of DDD where numerous changes within the cellular and extracellular compartments are related to the degenerative cascade. In DDD, fragmentation of small leucine-rich proteoglycans (SLRPs) such as decorin and biglycan occurs, alongside the loss of nucleus pulposus (NP) cell viability, downregulated synthesis of extracellular matrix proteins such as aggrecan and collagen type 2, increased expression of matrix-degrading proteins such as MMP-3, -13 and ADAMTS-4 (and the associated loss of proteoglycans), increased expression of pro-inflammatory cytokines and a generalized catabolic milieu [7,8]. The decrease in the levels of collagen type 2 and aggrecan alongside associated gross changes in the IVD phenotype—such as a loss of disc height, disorganized NP, impaired biomechanical properties, increase in pain-related proteins expressed within the disc (principally the annulus fibrosus), and aberrant Modic changes in the subchondral bone of the vertebral endplates—define degenerative disc disease [9,10,11,12,13,14,15,16,17]. It is unclear if there is a single precipitating cause of DDD (perhaps aside from a traumatic lesion) and it is most likely that the pivotal events discussed above occur simultaneously in some cases and as downstream changes in others. These complex physiological events present considerable challenges to the development of novel therapeutics, which will need to surmount the salient cellular/molecular changes unique to DDD.

## 2. Tissue Milieu Challenges

The IVD is a hypoxic, ischemic, avascular and immune-privileged tissue compartment that relies upon the diffusion of gases and nutrients through the capillary buds that envelop the vertebral cartilage endplates for cellular nutrition and gas exchange.

### 2.1. Vertebral Endplates 

Contemporary theories suggest that calcification of the tiny capillary buds that are embedded within the vertebral endplates, leads to a loss in nutrient supply and waste evacuation that is inextricably linked to DDD [16,17,18]. These endplate and diffusion changes result in the development of a pro-catabolic milieu with lowered concentrations of oxygen and glucose and elevated levels of lactic and propionic acid and carbon dioxide [4,5,6,11,14,16,19]. It is unknown whether the endplate changes and capillary calcification are an inciting event or whether they accompany other changes, such as increased pro-inflammatory cytokine expression within the NP, inflammation associated with an increase in annulus fibrosus neuropeptide/neurotrophin expression, loss of NP cell viability, and sub-chondral Modic changes at the vertebral body/endplate interface [11,13,16,17,18,20,21]. In addition to the loss of cell viability within the NP, there is a shift to the production of collagen type 1 in the nucleus and a loss of collagen type 2, as well as aggrecan, all of which contribute to the overall degenerative process and a shift in phenotype to that of a degenerative disc. Commensurate with these tissue changes is a loss of disc height and compromised biomechanics, all of which result in symptoms of pain and disability [22]. An effective biological agent to address DDD would need to convey its therapeutic effect within this harsh tissue compartment.

### 2.2. Molecular Mechanisms and DDD

Tissue changes inherent to the progression of DDD are clearly driven by changes in molecular regulation of the tissues (nucleus pulposus, annulus fibrosus, vertebral endplates) involved. To date, quite a number of published accounts involving in vitro and some in vivo molecular assays have illustrated the importance of increased expression of pro-inflammatory cytokines such as MMP-13, MMP-3, ADAMTS 4/5, and various collagenases and have clarified the important role that inflammation plays in the progression of DDD [7,8,13,14,16,23,24,25].

## 3. Degeneration vs. Symptoms

The cellular and molecular aspects of DDD have been extensively studied and although many aspects remain unknown, there is a well-characterized change in IVD phenotype from healthy to degenerative with impaired diffusion through endplates, development of a pro-cell death milieu within the NP, loss of structural integrity and compromised biomechanics. However, it is vital to be aware that the main reason for patient visits to their providers is back/neck pain (with or without radiation of pain to the leg/arm); therefore, any effective therapy must also have utility in the mediation of DDD-related pain. Neurological mechanisms involved in the development of IVD tissue-specific pain have been identified that involve inflammation-associated changes in innervation of the IVD that promote the expression of neuropeptides such as substance P, calcitonin gene-related peptide and neurotrophins such as brain-derived neurotrophic factor (BDNF) and its receptor TrkB, and nerve growth factor (NGF) and its receptor TrkA [23,25,26,27]. In particular, a rodent study involving painful whole-body vibration conclusively showed the development of neurotrophin proteins within cervical spine IVDs that was not present in controls [25]. Therefore, biological therapy would be faced with overcoming/mediating these tissue changes that are associated with discogenic pain.

## 4. Degenerative Disc Disease-Related Pain, Pain Assessments and Imaging

### 4.1. Origin of Disc Pain 

There are many ways that the degenerative disc can induce pain, including local structural damage and associated inflammation (disc herniation and irritated local structures), degeneration-induced osteophyte nerve irritation, inflammatory-related Modic changes in the vertebral endplates, compromised biomechanical properties, and neurogenically induced pain pathways that involve neurotrophin and neuropeptide expression in the IVD itself and/or local dorsal root ganglia [5,6,22,25,27,28,29,30]. Furthermore, a number of these changes can co-exist leading to complex pain pathways that develop because of the contribution of one or more of these pathologies. Because multiple aspects of degenerative-related IVD changes can contribute to pain, it is difficult to develop a single application that may mitigate some of these tissue changes and the development of pain. In addition, the development of pain-related behaviors can add another dimension to the development of spinal pain that is associated with DDD.

### 4.2. Pain Assessments

Psychogenic determinants of back/neck pain are variable and beyond the scope of this report, but considerations of contributors to pain and disability that are not clearly associated with pathology can add a confusing element to the diagnosis and treatment of patients with complex DDD-related pain behaviors. In particular, a biological intervention in the case of back/neck pain that is not performed in the case of a well-defined pain-generating tissue source could result in therapeutic ’failure’ and confuse metrics designed to evaluate the efficacy of the therapy [28]. Pain of psychologically related origin has been associated with depression, socio-economic status, certain ethnicities, substance abuse, employment status and third-party compensation (such as workers’ compensation and motor vehicle settlement status) and as such must be considered in the treatment of such patients [31,32,33,34,35]. Amongst those who specialize in pain medicine, descriptors such as “psychologically augmented pain” have been suggested as replacements for past references to ‘psychosomatic’ or ‘psychogenic’ pain that more recently, have been considered to be misleading [36]. Further, terminology such as “excessive illness behavior” may be preferred to “conscious or unconscious exaggeration of pain” [36]. There are many patient-centric metrics with which to classify pain and pain behavior such as the Oswestry disability questionnaire, visual analogue pain scales, and various psychological questionnaires that also encompass health-related quality of life metrics such as the SF-36 questionnaire, PROMIS and others [37]. These questionnaires are of course subject to patient-reporting bias but do provide some insight into possible psychologically augmented pain aspects to patient-reported quality of life that may help clinicians to develop the best treatment. In fact, it is important for clinicians to avoid overly ‘silo-ing’ the patient into only treating a tissue condition when the psyche may be an important part of the patient’s pain syndrome [28,34]. The prime objective of any regenerative therapy must make a significant impact on pain in addition to whatever cellular molecular changes that may occur, and it is important to be cognizant of the nexus between tissue pathology and pain; this issue makes the quantification of pain an important element. The objective determination of a pain source that may be treated must be judged on balance with whatever ‘psychologically augmented pain’ may be present to obtain the best result. It is important to be mindful when approaching a patient suffering from chronic spinal pain that it is likely impossible to discern the learned pain behaviors developed because of living with chronic pain. Therefore, despite the tissue source of pain that may be present, non-organic pain behaviors need to be recognized and accounted for within any treatment plan.

### 4.3. Imaging Modalities 

To date, investigations of DDD and its associated pain have mainly focused upon pain thought to emanate from the IVD itself—principally of the nucleus and annulus as determined by provocative discography (PD). However, whatever utility PD may have, it is not capable of discriminating pain from other IVD structures such as the vertebral endplate or sub-chondral bone because it is dependent on IVD pressurization. Emerging technologies such as MRI spectroscopy and ultrashort echo time (UTE) MRI may offer revolutionary changes in the ability to not only image the IVD and vertebral endplates, but also non-invasively quantify pain associated with these structures. UTE MRI can depict improved contrast of vertebral endplate tissues and may assist in the quantitation of pain emanating from these the tissues [38,39]. MRI spectroscopy (MRS) is a technique under investigation that can non-invasively assess the levels of certain metabolites within the IVD NP such as lactate and propionic acid that detect pain determinants within the IVD NP, as well as collagen and proteoglycan ratios that change with progressive degenerative disease [40,41,42]. Clinical studies have shown a strong correlation of MRS with positive provocative discography (up to 93% in non-herniated discs) but with the added advantage of being completely non-invasive [40]. The degenerative disc develops a change in the proteoglycan/collagen ratio as well as the quantity of molecules associated with pain including lactic acid and propionic acid thus achieving a quantifiable score with which to compare signals from adjacent discs and determine the symptomatic disc to guide treatment [40]. The evolution of MRS as well as UTE may offer the clinician the ability to determine pain emanating from a specific IVD and/or the vertebral endplate in a fashion currently impossible to assess. The ability to more discriminately and accurately identify the painful tissue compartment will greatly facilitate treatment options and avoid misguided interventions [40]. It cannot be over emphasized that the ability to objectively deduce determinants of the generators of pain will greatly assist the development of novel technologies that may be able to reduce reliance upon patient-reported outcomes. Pain is a subjective experience; however, to perform a molecular/cellular therapy on a patient living with discogenic pain, it is essential to determine, as accurately as possible, what and where the pain generator is and to be able to monitor the patient’s response to therapy as quantitatively as possible. Self-reported pain measurements are important; however, self-reported metrics have been reported to vary with time, experience, genetics and other non-objective aspects [43,44].

## 5. Challenges to Regenerative Therapy

As described earlier, the degenerative IVD develops a hostile milieu due to its diminished nutrient status, compromised gas and solute transport, changes in cellular and extracellular matrix components and altered phenotype. Such cellular and molecular changes present formidable obstacles to the success of any biological therapy. To date, a few attempts at biological therapy to address DDD have been made in pre-clinical models, with some in clinical trials. A review of clinicaltrials.gov using the search term ‘degenerative disc disease-injectable’ yielded 65 studies listed with varying statuses from incomplete, withdrawn, recruiting or no results posted. One trial study (study identifier NCT02320019) named “Clinical trial of YH14618 in patients with degenerative disc disease Yuhan Corporation” was completed on 8 November 2016; however, no results are posted. The treatment used was a TGF-β1 inhibitor thought to resolve aberrant TGF-β1 signaling in patients with DDD. The results on the safety and tolerability study of this injectable were published and it was concluded that the intervention was safe and well tolerated, but no difference in disc height index or MRI grade was reported after 48 weeks of follow up. However, there were variable improvements in VAS scores over the 48-week study [45].

Another potential biologic, SM04690 (a Wnt inhibitor), showed a reduction in disease progression of DDD in a rat model, suggesting that it could have disease-modifying potential effects. However, the clinicaltrials.gov website reports that a small study of this treatment in six humans was terminated for business reasons, with no results provided.

With respect to cellular therapies, a few clinical trials using various forms of stem or engineered IVD NP cells have been undertaken. One such study involved the injection of 6 or 18 × 10^6^ mesenchymal precursor cells coupled with hyaluronic acid, hyaluronic acid vehicle control or a saline (placebo injection). The results of this study showed no change in the modified Pfirrmann score; however, there were some statistically significant changes in the visual analogue pain scale (VAS) and Oswestry disability index (ODI) that showed a superior result of cell transplants compared to controls at 36 months [45]. The results of this and other clinical trials highlight some of the challenges involved with molecular or cellular-based therapies and the lack of solid, quantifiable data without a reliance upon variable patient-reported outcomes.

There are other studies involving small molecules, proteins and cell-based approaches listed on the clinicaltrials.gov website, and the reader is referred to either https://clinicaltrials.gov/search?cond=Degenerative%20Disc%20Disease&intr=injectable (accessed on 12 September 2024) or https://clinicaltrials.gov/search?cond=Degenerative%20Disc%20Disease&term=Cell%20Therapy&intr=injectable (accessed on 12 September 2024) for a listing of other trials and interventions that are too numerous to list here. An important aspect is that some possible therapies have been or are currently under clinical trial investigation, but convincing results are yet to be determined. The following sections discuss these possible approaches.

### 5.1. Cellular Replacement

DDD leads to changes within the IVD NP milieu (accelerated hypoxia, inflammation, and catabolic processes that overwhelm anabolic repair) that in part, lead to accelerated cell death [24,46]. Therefore, it might seem an attractive option to investigate cellular replacement to ameliorate the loss of ECM and cellular secreted factors. Cellular replacement may be based upon endogenous IVD NP cells from donors or stem/progenitor cells (from a variety of sources such as bone marrow, induced pluripotential stem cells, or umbilical-cord-derived stem cells) [47,48,49,50,51]. Several obstacles have created difficulties with this form of therapy, leading to the following questions: What is the optimum number of cells that can be safely transplanted? How long will the transplants survive? Do the transplanted cells engraft into the IVD NP, and do they synthesize de novo ECM? How can the diminished nutrient supply and solute transport within the degenerative IVD NP sustain transplanted cells, particularly if the transplants are in the millions of cells? Is there a possible risk of immune reaction to the transplant of foreign cells? What implications might there be in the case necessary repeated treatment? Furthermore, there has been scant attention paid to the effect of cell transplantation upon the ability of transplanted mesenchymal stem cells (MSCs), for example, and their effects upon sensory nerve interaction with the transplanted milieu. The potential implications of the effects on pain after cell transplantation are highlighted by observations that transplanted MSCs may upregulate nerve growth factor and its receptors [52]. Therefore, an unresolved question with respect to cellular transplant is whether it may lead to unintended aggravation of pain, as conveyed by the degenerative disc that may be undergoing increased pain receptor expression [52]. Several clinical studies have been performed using bone-marrow-derived mesenchymal cells and engineered stem cells to overexpress particular growth factors and cytokines and even utilize novel matrices [53]. However, most of these studies are hampered by small sample size, selection bias, short follow up and a lack of control groups [51]. One study by Henriksson et al., examined the engraftment of transplanted stem cells in four patients after spinal fusion [54]. The tissue samples were obtained due to the patients electing to have spinal fusion 8 months post-transplantation in three cases and at 28 months in the fourth case. The tissues examined showed some engraftment of the transplanted cells; however, this attempt must be considered a failure as all participants elected to have spinal fusion surgery as soon as 8 months post-operation. Furthermore, many studies suggest that the main mechanism(s) involved in the therapeutic benefit derived from stem cell transplants is associated with the secretory products of the stem cells themselves, such as immunomodulatory factors, growth factors, and anti-inflammatory cytokines [51,53,55]. The notion that the main therapeutic benefit derived from cellular transplant may be associated with the cellular secretome raises the question of why are cells transplanted rather than delivering the therapeutic molecules themselves? There are also significant regulatory hurdles involved with cellular transplant and important ethical considerations such as, in the case of embryonic stem cells, induced pluripotent stem cells (iPS cells) and the possibility of neoplastic change.

### 5.2. Regenerative Molecules

With respect to therapeutic molecules, numerous approaches have been undertaken to treat DDD using molecular methods including the application of members of the TGF-β superfamily of growth factors (TGF-β1, β2, β3, GDF-5, BMP-2, BMP-7/OP-1), CCN-2), and, in what may seem like a paradox, even using various TGF-β inhibitors [29,56,57,58,59]. The use of these various proteins has originated in many cases, from developmental studies of the skeletal system and they have been variably delivered to the disc by direct injection of the proteins themselves, via gene therapy. Other potential biological therapies that have been studied in pre-clinical evaluation include IL-6 receptor antibodies, Lovastatin, Cox-2 inhibitors such as Celecoxib, the EGF receptor inhibitor, LINK-N, and various statin drugs [60]. Although these therapeutics show some utility in pre-clinical animal studies, there has yet to be a strong demonstration by any of the proteins listed above that demonstrates longevity of effect or robust anti-catabolic and pro-anabolic tissue level changes that would realistically define regeneration. One method to deliver therapeutic proteins to the IVD is via gene therapy, whereby various forms of viral vector can be used to deliver the therapeutic protein. However, despite the potential to utilize the native cells of the IVD NP to be engineered to upregulate the secretion of desired proteins, safety considerations have arisen with respect to gene therapy in the IVD and its possible unwanted effects [61,62]. Also, most, if not all, of the studies involving these proteins fail to demonstrate utility in addressing the pivotal element of DDD, which has a positive effect upon pain.

Combinatorial therapy using recombinant proteins may take advantage of common signaling properties and induced tissue level changes that offer superior benefits to single proteins only. With respect to combinatorial therapy, published accounts by Matta et al. based upon notochordal cell secretome analysis, have demonstrated that a single injection into a needle-puncture-injury-induced DDD in a chondrodystrophic canine model with a combination of TGF-β1 + CCN-2 yielded significant pro-anabolic and anti-catabolic effects, preserved disc height, and improved biomechanical properties compared to vehicle injection that persisted for at least to 14 weeks post-injection [63,64,65]. This work originated from a comprehensive analysis of the secretome of the notochordal cell-rich non-chondrodystrophic canine IVD. Non-chondrodystrophic canines (outbred dogs, Greyhounds, Labrador Retrievers and German Sheppards) retain the population of notochordal cells within the nucleus pulposus until much later in life, retain a more gelatinous nucleus pulposus, and are not known to develop spontaneous degenerative disc disease. However, chondrodystrophic dogs such as Beagles, Dachshunds, Corgis, Pekingese, and Bassett Hounds, lose their nucleus pulposus notochordal cells in favor of chondrocyte-like cells and develop a more fibrocartilaginous nucleus pulposus akin to humans and are well reported to suffer from early degenerative disc disease (Figure 1) [66,67,68,69,70]. Furthermore, and significantly with respect to the clinical challenge of mediating pain, it was also shown that this single combinatorial injection suppressed the inflammation and the expression of pain-related neurotrophins and neuropeptides within the treated discs to baseline healthy levels, whereas saline injections yielded significantly elevated levels of these proteins within the annulus fibrosus of the IVD [30]. Although this published account did not provide specific pain-related data (due to limitations in pain evaluation in large animal models), it does indicate the capacity of biological therapy to affect tissue mechanisms related to pain of discogenic origin. This aspect is important when considering the impact of biological therapy with respect to pain, as discussed in the prior section concerning cellular transplant. There are a host of pro-inflammatory and pro-catabolic mediators that arise within the degenerative disc nucleus pulposus, and an effective therapy must be able to counter these pro-catabolic processes and also provide a satisfactory pro-anabolic response (Figure 2).

To maximize the beneficial aspects of therapeutic molecule delivery, the use of encapsulation strategies or associated biomaterials is important to consider. The use of biomaterials/delivery systems requires consideration of aspects such as iso-electric point and molecular weight because these elements can factor into the interplay between the therapeutic molecule (such as a growth factor) and the delivery system itself. The rate of exchange of a therapeutic molecule within a delivery system (microencapsulation, hydrogels or other matrices) may figure prominently in the ability to deliver a stable therapeutic product over time at the appropriate concentrations and diffusion schedules. Such integration and timely delivery of therapeutic proteins are critical components to consider within the overall realm of tissue engineering. The review by Cabaleero Aguilar provides a very good summary of growth factor delivery systems and their possibilities [71].

### 5.3. Extracellular Matrix-Related Therapy

As described above, the chondrodystrophic canine (CC) is susceptible to DDD beginning at 3 years of age, whereas the non-chondrodystrophic canine (NCC) is remarkably resistant to developing DDD until later in age, if at all [69,72,73,74]. Accordingly, there has been considerable interest in determining the nature of mechanisms/factors that may be responsible for such protection from DDD. From the perspective of modeling therapies to address human degenerative disc disease, the choice of animal model is vital, and studies by Miyazaki et al. and Alini et al. provide valuable insight in this regard [75,76]. Groups interested in the possible contribution of the nucleus pulposus of NCC animals to anti-degenerative strategies have demonstrated some utility by exploiting processed/lyophilized extracellular matrix (ECM) from nucleus pulposus of notochordal cell-rich porcine NPs [77,78]. Although such protein-rich ECM may contain bioavailable factors produced by notochordal cells (that have been shown to have beneficial effects in other DDD-related studies, see previous section), difficulties with regulatory hurdles in the translation of using porcine proteins and controlling dosing and characterization of complex ingredients may limit a path to the clinic [78]. Furthermore, because the notochordal cell-rich IVD is protective of the IVD, it may be better to deliver the essential proteins directly rather than rely upon an incompletely characterized processed porcine NP ECM.

### 5.4. Breakthrough Technologies

With emerging imaging technologies such as MRI spectroscopy and UTE MRI, it may be possible to more accurately determine the source of pain in a patient suffering from DDD, a vital determination through which a potential breakthrough therapy may be instituted. For example, currently, the only method to determine pain in a patient with DDD is using provocative discography. A few clinical trials have used this approach that required multiple disc punctures, which may lead to advanced DDD, and clinical trials have yet to produce satisfactory results using a sufficient sample size and duration of effect. A non-invasive method to determine the tissue target for the putative therapy will minimize tissue damage and allow a better choice of therapy. For example, a treatment that may address DDD within the NP may not be suitable for pain of endplate origin. In contrast, it may be that IVD NP treatment could have an indirect effect on the vertebral endplate, and therapy ostensibly developed to treat the IVD NP could possibly have beneficial effects on the endplate. In future studies, these new emerging imaging modalities may be able to track/determine what tissue level changes occur with a given therapy with much more precision. Regardless of the diagnostic methods, therapy that can mediate the progression of DDD and have a meaningful impact upon pain itself remains to be determined. The publications by Matta et al. may offer some promise for such an effect; however, these efforts remain in the pre-clinical phase and any benefit for humans is yet to be determined [30,65]. Whether the exciting pre-clinical treatments that show promise in animal models will translate to humans and how the financial and regulatory burden of translating these potential therapies can be surmounted to bring novel, cellular/molecular therapies to the clinic is unknown. Nonetheless, over recent years, advances in imaging and understanding of disc-related pain mechanisms have clearly led to improvements in our approach to treating DDD.

## Figures and Tables

**Figure 1 cells-13-02103-f001:**
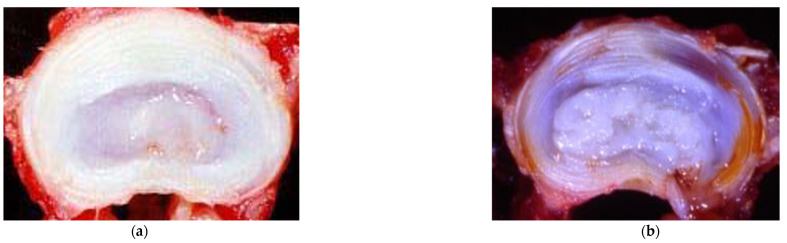
Images of non-chondrodystrophic and chondrodystrophic canine IVDs. (**a**) Non-chondrodystrophic canine IVD sourced from 3-year-old outbred canine. (**b**) 3-year-old chondrodystrophic canine (Beagle) IVD. Note the gelatinous appearance of the non-chondrodystrophic canine compared to the fibrocartilaginous appearance of the chondrodystrophic canine.

**Figure 2 cells-13-02103-f002:**
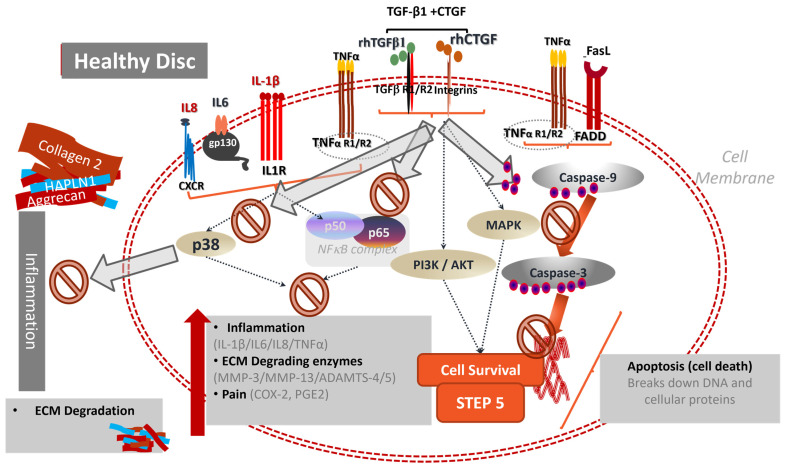
Schematic of pro-inflammatory intracellular signaling induced by IL-1β, TNF-α, IL-6, IL-8 and Fas Ligand and the anti-inflammatory and pro-anabolic effects induced by the intra-discal injection of TGF-β1 and CTGF. The pro-inflammatory cytokines, principally IL-1β, induce downstream activation of P38 MAPK and translocation of the NFκβ complex to the nucleus that in turn increase inflammation (IL-6, IL-8, and TNF-α) and extracellular matrix remodeling enzymes such as MMP-3, -13, ADAMTS 4/5. The addition of TGF-β and CTGF suppresses the expression of these inflammatory proteins and suppresses cell death via suppression of the activation of Caspase-3.

## Data Availability

Not applicable.

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
