# Peer review of "Intervertebral Disc Degeneration and Regeneration: New Molecular Mechanisms and Therapeutics: Obstacles and Potential Breakthrough Technologies"

_cells, 2024, doi:10.3390/cells13242103_

Round 1

Reviewer 1 Report

Comments and Suggestions for Authors

The objective of this study is to provide a comprehensive review of the current therapeutic approaches for intervertebral disc (IVD) and degenerative disc disease (DDD), along with an analysis of the challenges and opportunities for novel treatment strategies. The review is well written and concise. The review succinctly summarises the key points of the disease. However, a few modifications are required:

- minor corrections to English are necessary (e.g. line 22).

- the title is not optimally written, with double ":"

- some references are missing. For instance, in the section "cellular replacement" (line 213), "extracellular matrix-related therapy" (line 264). Additionally, the section on "breakthrough technologies" is devoid of references.

Comments on the Quality of English Language

Minor corrections to English are necessary

Author Response

Thank you for your valuable suggestions.  We have modified the English where necessary and hope that these changes are now sufficient.  We have reviewed the references and corrected them where necessary.  The section on “breakthrough technologies has been extensively revised with appropriate references. 

Reviewer 2 Report

Comments and Suggestions for Authors
  • Figures and Schemes: The manuscript lacks figures and schemes, which would greatly enhance the clarity and impact of the content presented. Consider incorporating relevant visual elements, such as diagrams, flowcharts, or illustrative figures, to better support and communicate key points.

  • References: Many of the references cited in the manuscript are outdated, and the review would benefit from the inclusion of more recent studies. To strengthen the manuscript, I recommend updating the references by including the latest research findings. For example, consulting ClinicalTrials.gov or similar databases for ongoing or recent clinical trials in this area could provide valuable insights and help highlight emerging trends and topics.

Author Response

Thank you for your insightful and helpful comments.  We agree that some images may be helpful and have endeavored to provide them including a schematic regarding the effect of pro-inflammatory cytokines on cell processes and possible therapeutic mitigation.  We have added many newer references including extensive use of clinicaltrials.gov and have attached links to the select data bases for therapeutic molecule and cell-based approaches to help the reader.  We do feel that although some references may be a bit older, this does not disqualify their use-nonetheless, adding the newer ones we feel, will make the manuscript more timely, thank you for the suggestion.

Reviewer 3 Report

Comments and Suggestions for Authors

he review addresses the current need to address an “old” problem, i.e., IVD degeneration. 

I think the review is well-written and provides stimulus for future therapies. 

I had the following comments to improve the article: 

Could an example of 2-3 dog breeds be given for both the chondrodystrophic and the non-chondrodystrophic dogs? It is easier than understanding and visualizing which species the authors are talking about. Also, it should be recapitulated for the non-experts of dog breeds which line keeps the notochordal cells (NC) throughout and which dog lines are losing these, similar to the human situation. 

Further essential differences between the species keeping the NCs and those that lose the NCs should be made, like the consistency of the NP, which is much more jelly-like in non-chondrodystrophic dogs. 

Maybe the review could also mention the role of the cell densities like described here. 

1. Miyazaki T, Kobayashi S, Takeno K, Meir A, Urban J, Baba H (2009) A Phenotypic Comparison of Proteoglycan Production of Intervertebral Disc Cells Isolated from Rats, Rabbits, and Bovine Tails; Which Animal Model is Most Suitable to Study Tissue Engineering and Biological Repair of Human Disc Disorders? Tissue Eng Part A 15(12):3835-3846 https://doi.org/10.1089/ten.tea.2009.0250

And here: 

2. Maroudas A, Stockwell RA, Nachemson A, Urban J (1975) Factors involved in the nutrition of the human lumbar intervertebral disc: cellularity and diffusion of glucose in vitro. J Anat 120(1):113-130

 The article does not visualize at all what it is talking about. Maybe an overview figure could be shown how biologics, EC-based approaches or other regenerative approaches could be taken.

Author Response

We have taken your valuable and insightful comments into consideration and agree that examples of the two dog breeds requires more description, and we have done so.  We have also highlighted the importance of the appropriate animal model to study degenerative disc disease, have added the Miyazaki et al and the Alini et al references as well as figures showing the difference in phenotype between chondrodystrophic and non-chondrodystrophic canine IVDs. 

Round 2

Reviewer 2 Report

Comments and Suggestions for Authors

The authors have thoroughly addressed the comments and suggestions provided during the review process. The revised manuscript demonstrates significant improvements, and the authors have adequately clarified and resolved the issues previously raised. The study is now well-structured

Based on the comprehensive revisions and the quality of the manuscript, I recommend its acceptance for publication.